# A Gelatin Hydrogel Nonwoven Fabric Enhances Subcutaneous Islet Engraftment in Rats

**DOI:** 10.3390/cells13010051

**Published:** 2023-12-26

**Authors:** Ryusuke Saito, Akiko Inagaki, Yasuhiro Nakamura, Takehiro Imura, Norifumi Kanai, Hiroaki Mitsugashira, Yukiko Endo Kumata, Takumi Katano, Shoki Suzuki, Kazuaki Tokodai, Takashi Kamei, Michiaki Unno, Kimiko Watanabe, Yasuhiko Tabata, Masafumi Goto

**Affiliations:** 1Department of Surgery, Tohoku University Graduate School of Medicine, Sendai 980-0872, Japan; ryusuke009009@gmail.com (R.S.);; 2Division of Transplantation and Regenerative Medicine, Tohoku University Graduate School of Medicine, Sendai 980-8575, Japan; 3Division of Pathology, Graduate School of Medicine, Tohoku Medical and Pharmaceutical University, Sendai 983-8536, Japan; 4Laboratory of Biomaterials, Department of Regeneration Science and Engineering, Institute for Life and Medical Sciences (LiMe), Kyoto University, Kyoto 606-8507, Japan

**Keywords:** islets, gelatin hydrogel nonwoven fabrics, subcutaneous transplantation, extracellular matrix, neovascularization, rats

## Abstract

Although subcutaneous islet transplantation has many advantages, the subcutaneous space is poor in vessels and transplant efficiency is still low in animal models, except in mice. Subcutaneous islet transplantation using a two-step approach has been proposed, in which a favorable cavity is first prepared using various materials, followed by islet transplantation into the preformed cavity. We previously reported the efficacy of pretreatment using gelatin hydrogel nonwoven fabric (GHNF), and the length of the pretreatment period influenced the results in a mouse model. We investigated whether the preimplantation of GHNF could improve the subcutaneous islet transplantation outcomes in a rat model. GHNF sheets sandwiching a silicone spacer (GHNF group) and silicone spacers without GHNF sheets (control group) were implanted into the subcutaneous space of recipients three weeks before islet transplantation, and diabetes was induced seven days before islet transplantation. Syngeneic islets were transplanted into the space where the silicone spacer was removed. Blood glucose levels, glucose tolerance, immunohistochemistry, and neovascularization were evaluated. The GHNF group showed significantly better blood glucose changes than the control group (*p* < 0.01). The cure rate was significantly higher in the GHNF group (*p* < 0.05). The number of vWF-positive vessels was significantly higher in the GHNF group (*p* < 0.01), and lectin angiography showed the same tendency (*p* < 0.05). The expression of laminin and collagen III around the transplanted islets was also higher in the GHNF group (*p* < 0.01). GHNF pretreatment was effective in a rat model, and the main mechanisms might be neovascularization and compensation of the extracellular matrices.

## 1. Introduction

The subcutaneous space has many advantages as an alternative transplant site for pancreatic islets and, because it is large enough to support a large number of islets, it is easily accessible, minimally invasive, easy to monitor [1], and/or the islet grafts can be removed if necessary [2,3]. However, the subcutaneous site is poorly vascularized, and it must be modified before transplantation in order to support islet grafts [4]. Various subcutaneous approaches have been developed for delivering grafts, including the use of devices [5,6], biomaterials [7,8], endothelial cells [9], and fibroblasts [10]. Although all these approaches have improved islet engraftment to some degree, a large number of islets are still necessary to achieve normoglycemia.

Alternative strategies for subcutaneous islet transplantation using a two-step approach have been proposed, in which a favorable cavity for transplantation is first prepared using various materials, followed by islet transplantation into the preformed cavity [11]. These approaches have several advantages: namely, new vessels for supporting islet grafts are induced in advance and foreign body responses to materials are reduced when the islets are transplanted. We recently reported that new vessels in the subcutaneous tissues continuously enriched until six weeks after the placement of materials [12]. We also demonstrated that inflammation in the subcutaneous space was extremely severe until two weeks after the implantation of the materials and that islet engraftment tended to be poor if the islets were transplanted during this period. These approaches efficiently deliver oxygen and nutrition to islet grafts and enable them to respond to hyperglycemia immediately, which is similar to the response observed when they are transplanted into the portal vein or the kidney subcapsular space [13,14].

We previously reported the efficacy of pretreatment with a recombinant peptide (RCP: alpha-1 sequence of recombinant collagen type I supplemented using a 12 RGD [Arg-Gly-Asp] motif 1 molecule) device for improving subcutaneous islet engraftment [15]. The main drawback of the RCP device is that it is barely absorbed in the body, and it must be removed at the time of transplantation. In this process, substantial amounts of constructed vessels and the extracellular matrix (ECM) are destroyed. Therefore, we alternatively used a bioabsorbable gelatin scaffold called the gelatin hydrogel nonwoven fabric (GHNF). GHNF was originally produced as a useful scaffold for various types of cell cultures [16,17]. It is known to degrade gradually and thereafter is absorbed in vivo and finally replaced by the host tissues; therefore, it can be an ideal pretreatment material for our purpose since there is no need for removal. In fact, we could effectively restore diabetic mice with a marginal dose of subcutaneously transplanted islets [12]. Moreover, the results of subcutaneous islet transplantation using GHNF combined with adipose tissue-derived stem cells were surprisingly superior to those of intraportal transplantation, which is the current standard procedure (manuscript submitted). However, these approaches have only been applied to a mouse model, and the efficacy of subcutaneous islet transplantation in animal models other than mice is still extremely low compared to intraportal transplantation [18].

Therefore, in the present study, we adapted the GHNF pretreatment method for a rat model, in which GHNF sheets sandwiching a silicone spacer were implanted in the subcutaneous space of Lewis rats for three weeks (GHNF was absorbed more rapidly in a rat model), followed by islet transplantation into the space where the silicone spacer was removed. We investigated whether the preimplantation of GHNF could induce efficient new vessels and thereby improve subcutaneous islet engraftment, even in a rat model.

## 2. Materials and Methods

### 2.1. Animals

All animals used in this study were handled in accordance with the Guide for the Care and Use of Laboratory Animals published by the National Institutes of Health [19]. All experimental procedures used in this study (protocol ID:2020 MdA-022) were approved by the Animal Experimental Committee of Tohoku University. Lewis rats were used as recipients and islet donors (8–10 weeks of age; Japan SLC Inc., Shizuoka, Japan). All surgical operations were performed under anesthesia, and efforts were made to minimize suffering.

### 2.2. The Induction and Diagnosis of Diabetes in the Recipients

Seventy mg/kg streptozotocin (STZ) (Sigma-Aldrich, Inc., St. Louis, MO, USA) was injected into the penile vein to induce the diabetes. Rats with non-fasting blood glucose levels ≥400 mg/dL on two consecutive measurements were determined to be diabetic. Blood glucose levels were measured every 3–4 days and recipients whose non-fasting blood glucose levels decreased less than 200 mg/dL on two consecutive measurements were determined to be cured.

### 2.3. Islet Isolation

Islet isolation and culture were performed as previously described [20]. Briefly, the papilla of Vater was clumped by the clip, then 10 milliliters of 1 mg/mL collagenase (Sigma type V; Sigma Chemicals, St. Louis, MO, USA) diluted with cold Hank’s balanced salt solution (HBSS) was injected into the common bile duct. The resected pancreas was incubated in a water bath at 37 °C for 12 min. The cell suspension was washed three times in cold HBSS and then centrifuged for 1 min. Density-gradient centrifugation was performed for 10 min using Histopaque-1119 (Sigma Diagnostics, St. Louis, MO, USA) and Lymphoprep™ (Axis-Shiled, Oslo, Norway) to isolate pancreatic islets. The islets were cultured overnight in Roswell Park Memorial Institute-1640 medium containing 5.5 mmol/L glucose and 10% fetal bovine serum at 37 °C in 5% CO_2_ and humidified air before transplantation.

### 2.4. Preparation of a Gelatin Hydrogel Nonwoven Fabric and Islet Transplantation

A gelatin hydrogel nonwoven fabric (GHNF; NIKKE MEDICAL Co., Ltd., Osaka, Japan) was prepared using a previously reported method [16] and processed into a circular sheet (22 mm diameter, 0.5 mm thickness). Two GHNF sheets sandwiching a silicone spacer (26 mm diameter, 0.5 mm thickness) were placed into the left dorsal subcutaneous space 3 weeks before islet implantation (GHNF group), and a silicone spacer alone was also arranged in the same way (control group). The duration of pretreatment was determined according to a previous report in mice by comparison with hematoxylin–eosin staining [12]. GHNF and silicone spacers were implanted into healthy rats and STZ was injected 7 days before islet transplantation.

After removing the silicone spacer, different numbers of syngeneic rat islets were transplanted using a gastight syringe (Hamilton Co., Reno, NV, USA) to determine the marginal dose of islet grafts using GHNF that normalizes the blood glucose levels (2,700–4,680 islet equivalents (IEQs); *n* = 11, 5,400–6,300 IEQs; *n* = 7, 7,200–9,360 IEQs; *n* = 8). For evaluating the impact of GHNF, 5,400 IEQs of islets were implanted into the pretreated space in the GHNF and control groups. The recipients were followed by measuring the non-fasting blood glucose levels every 3–4 days throughout the study period (60 days after islet transplantation).

### 2.5. Intravenous Glucose Tolerance Test

An intravenous glucose tolerance test (IVGTT) was performed 60–65 days after islet transplantation as described previously [21]. In brief, recipients that had fasted for 12 h were intravenously infused with D-glucose (3.0 g/kg), and the blood glucose concentrations were measured before and at 5, 10, 20, 30, 60, 90, and 120 min after the injection of glucose. A blood glucose curve was generated, and the area under the curve (AUC) was used for comparison purposes.

### 2.6. Immunohistochemical Analyses

Three weeks after the subcutaneous placement of the GHNF sheets before islet transplantation, recipient tissue at the site of subcutaneous implantation was obtained, then fixed with 4% paraformaldehyde, and embedded in paraffin for immunohistochemical staining. Seven days after transplantation, the tissue at the site of subcutaneous islet transplantation was also obtained. Immunohistochemical staining was performed using anti-insulin (ab181547; Abcam, Cambridge, UK), anti-von Willebrand factor (vWF) (ab6994; Abcam), anti-collagen III (ab7778; Abcam), anti-collagen IV (ab6586; Abcam), and anti-laminin (ab11575; Abcam) antibodies. The EnVision+ System-HRP-labelled polymer anti-rabbit (4003; DAKO, Glostrup, Denmark) was used as the secondary antibody. To evaluate neovascularization, vWF-positive vessels in the capsule touching the silicone spacer were counted. In collagen III, collagen IV, and laminin staining, “positive” was defined as marked immunopositivity that was detectable in the fibrous capsule around the islets. More than six sections for each animal from each experimental group (GHNF; *n* = 5, control; *n* = 5) before islet transplantation and 10 sections after islet transplantation (GHNF; *n* = 5, control; *n* = 5) were evaluated by an experienced specialist familiar with pathology using a blinded method.

### 2.7. Islet Evaluation

Seven days after islet implantation, the subcutaneous tissue surrounding the transplanted islets was procured and stained with an anti-insulin antibody. All insulin-positive islets were evaluated in both groups. The area of β cells was calculated using the ImageJ 1.54d software program (National Institutes of Health, Bethesda, MD, USA) [22], and the area with a threshold of 0–70 was calculated as the insulin-positive area.

### 2.8. Lectin Angiography

Lectin angiography was carried out in the GHNF (*n* = 5) and control (*n* = 4) groups three weeks after GHNF placement [23]. The amount of tomato lectin was determined according to a previous report [24], but newly formed subcutaneous vessels could not be detected at this dose. Therefore, in this study, we used 3.75 mg/kg body weight of tomato lectin, which was three times larger than that used in the previous study. The rats were lightly sedated with isoflurane (Viatris Inc., Tokyo, Japan), and DyLight^®^ 488 Lycopersican esculentum agglutinin (LEA, tomato lectin; Vector Labs, Burlingame, CA, USA) was injected intravascularly via the penile vein. Five minutes after tomato lectin injection, the animals were perfused through the heart using an electric pump connected to a 24-G needle set at a pumping rate of 5 mL/min. Na-PO4 buffered 4% paraformaldehyde was used as the perfusion fluid. Following perfusion fixation, subcutaneous tissues at the pretreatment site were procured and placed in 4% paraformaldehyde and stored overnight in the dark at 4 °C. The following day, the tissues were washed twice with PBS and transferred to a solution of 10% sucrose in PBS. Six hours later, the tissues were transferred to 30% sucrose in PBS and stored in the dark at 4 °C for one day.

The tissues were cut using a microtome into about 50 μm thickness and then were mounted on slides, and coverslipped. Care was taken to protect the tissues from exposure to overhead room fluorescent light as much as possible. The sections were examined under a confocal microscope (LSM780; Carl-Zeiss, Oberkochen, Germany). The same controls for brightness and exposure times were used among the groups to maintain consistency. The images were imported into a Zeiss IMARIS (Carl-Zeiss, Germany) to perform vessel volumetry and analysis. The vascular volume in the subcutaneous capsules around the silicone spacer or GHNF was calculated. To determine the density of blood vessels, the capsular volume was calculated, and the vascular volume was divided by the vascular volume. The brightness value of the vascular area was defined as >20, and the minimum volume was defined as <5,000 μm^3^ to extract artifacts.

### 2.9. Statistical Analyses

All data are expressed as the mean ± standard error. All statistical analyses were performed using the JMP Pro 16 software program (SAS Institute Inc., Cary, NC, USA). Changes in the blood glucose levels and IVGTT were analyzed using a mixed-effect model analysis. The area under the curve (AUC) of the IVGTT, number of vWF-positive vessels, vascular density, and area of β cells were analyzed using Student’s *t*-test between the groups. The immunopositivity rate in the ECM evaluation was analyzed using Pearson’s chi-squared test. A log-rank test was used for analyzing Kaplan–Meier curves. *p* values less than 0.05 were regarded as a statistical significance.

## 3. Results

### 3.1. Time-Course Changes of GHNF Absorption under the Subcutaneous Space

After pretreatment with GHNF and a silicone spacer, subcutaneous tissues, including GHNF, were retrieved and stained with hematoxylin and eosin. GHNF was absorbed, and its volume gradually diminished. At four weeks after implantation, GHNF was completely absorbed and could not be observed (Figure 1A). At three weeks after implantation, the thickness of the capsule touching the silicone spacer in the GHNF group seemed to be thicker than that in the control group. The average area of the capsule in the GHNF group (observed at ×100 magnification) was significantly larger than that in the control group (GHNF: 338,209 ± 8,691 vs. control: 206,453 ± 7,472 μm^2^, *p* < 0.01) (Figure 1B).

### 3.2. The Comparison of Islet Engraftment after Marginal Islet Mass Transplantation between the GHNF and Control Groups

To determine the marginal dose of islet grafts in the GHNF group, different numbers of islets were transplanted into the pretreated space (2,700–4,680 IEQs; *n* = 11, 5,400–6,300 IEQs; *n* = 7, 7,200–9,360 IEQs; *n* = 8). As expected, the cure rate increased dose-dependently (2,700–4,680 IEQs; 9.1%, 5,400–6,300 IEQs; 42%, 7,200–9,360 IEQs; 75%, 2,700–4,680 IEQs vs. 7,200–9,360 IEQs, *p* < 0.05) (Figure 2A). According to these results, we determined the marginal dose of this model as 5400 IEQs and used this dose for further experiments. The blood glucose levels after islet transplantation in the GHNF group (*n* = 8) were significantly higher than those in the control group (*n* = 8) (*p* < 0.01) (Figure 2B). The cure rate of the diabetic rats 60 days after islet transplantation in the GHNF group was significantly higher than that in the control group (GHNF: 62.5% [5/8] vs. control: 12.5% [1/8], *p* < 0.05) (Figure 2C).

### 3.3. Intravenous Glucose Tolerance Test

Although the blood glucose changes in the IVGTT did not differ between the two groups (*p* = 0.16) as a whole, there were significant differences at 60 and 90 min after glucose injection (*p* < 0.05) (Figure 3A). Although the difference was not statistically significant, the AUC in the GHNF group was lower than that in the control group (GHNF: 31,257 ± 2,159 vs. control: 38,994 ± 3,252, *p* = 0.11) (Figure 3B).

### 3.4. Immunohistochemical Analyses

The number of vWF-positive vessels in the capsule touching the silicone spacer was counted and divided by the capsular area to examine neovascularization prior to islet transplantation (Figure 4A). The number of vWF-positive vessels in the GHNF group was significantly higher than those in the control group (*p* < 0.01) (Figure 4B). To examine the effect of GHNF pretreatment on the ECM components in subcutaneous capsules, the proportion of immunopositive sections in terms of collagen III, collagen IV, and laminin in the fibrous capsule around the islet grafts was evaluated (Figure 5A). Although there was no collagen IV positivity in either group, the rates of collagen III and laminin positivity in the GHNF group were significantly higher than those in the control group (*p* < 0.01) (Figure 5B).

### 3.5. Islet Evaluation

All insulin-positive islets in a pathological specimen obtained one week after islet transplantation were evaluated (GHNF, *n* = 60; control, *n* = 54). The average area of β cells in the GHNF group was significantly larger than that in the control group (GHNF: 97,486 ± 4,214 vs. control: 64,022 ± 4,954 μm^2^, respectively, *p* < 0.01) (Figure 6).

### 3.6. Quantification of the Vascular Volume Using Lectin Angiography

Lectin angiography revealed that the blood vessel density in the GHNF group was significantly higher than that in the control group (GHNF: 0.30 ± 0.032 vs. control: 0.19 ± 0.038, *p* < 0.05) (Figure 7A,B).

## 4. Discussion

The present study clearly showed that pretreatment with GHNF improved subcutaneous islet engraftment even in a rat model. Unlike in a mouse model, GHNF was completely absorbed over three weeks after implantation in a rat model. Three weeks after GHNF pretreatment, only a limited amount of GHNF remained in the subcutaneous capsules, but inflammation based on the foreign body responses to GHNF had already disappeared. In a rat model, GHNF preimplantation induced sufficient new vessels in the subcutaneous space compared to the control group (Figure 4B and Figure 7B). In addition, it enhanced ECM compensation surrounding the subcutaneously transplanted islets (Figure 5B).

The effectiveness of subcutaneous islet transplantation is extremely low, most likely due to the low concentration of oxygen and/or nutrition, which is based on poor vascularization in the subcutaneous space [25,26]. Therefore, optimization of the subcutaneous environment before islet transplantation is logical and it should be useful to improve the results of subcutaneous islet transplantation. In this process, the duration of pretreatment is of great importance for islet engraftment. Patikova et al. reported that the ideal timing of islet transplantation, when the capillary network and collagen synthesis are maximized, is important for islet engraftment [27]. We also reported that the length of the GHNF-pretreatment period had a strong impact on the results of subcutaneous islet transplantation and that six weeks was the best duration when neovascularization was maximized and inflammatory reactions were minimized in a mouse model [12]. However, the extent of absorption of GHNF varies among animal species. In our preliminary studies, GHNF was completely absorbed four weeks after implantation in a rat model added (Figure 1A). Therefore, we next compared the remnant volume of GHNF using HE staining in mouse and rat models. According to these evaluations, three weeks in a rat model, which corresponded to six weeks in a mouse model, was considered the best duration of pretreatment. In the present study, GHNF sheets were implanted in the subcutaneous space three weeks prior to islet transplantation, and islets were transplanted into the space where the silicone spacer was removed. If GHNF is applied to other species, including humans, observation of GHNF remnants by HE staining would thus be helpful in determining the ideal duration of pretreatment.

During islet isolation, ECM on the islet basement membrane is intensively digested by cell dissociation enzymes [28]. However, collagen III has been reported to improve islet function by protecting against apoptosis and increasing insulin secretion [29]. Likewise, laminin is also an important ECM that comprises 80% of the islet basement membrane and accelerates β-cell survival and insulin secretion [30,31]. Therefore, the compensation of the crucial ECM around the islet capsule, which is lost during islet isolation, is necessary for islet survival and function [14]. Considering the results of the present study, which showed that the immunopositive rate of laminin and collagen III in the GHNF group was significantly higher than that in the control group (Figure 5B), the ECM compensation around islets may be one of the crucial mechanisms of the GHNF effects in a rat model.

GHNF has been reported to be absorbed in vivo and replaced by recipient cells and ECM [17]. We previously reported that M2 macrophages, which have anti-inflammatory properties [32], accumulated between the gaps of GHNF [12]. The concentration of insulin-like growth factor-2, which maintains islet viability and has anti-apoptotic effects on islets [33,34], in the subcutaneous space, is high in the GHNF group in a mouse model [14]. Although we did not evaluate these issues in this study, these factors could be one possible explanation for the larger area of β cells observed in the GHNF group than in the control group (Figure 6).

Neovascularization in the subcutaneous space is undoubtedly an essential factor for obtaining a better and longer functioning of transplanted islets. The density of subcutaneous vessels was evaluated using vWF staining and lectin angiography. Lectin angiography can theoretically detect vessels less than 10 μm in diameter, and it enables three-dimensional evaluation of vessels in the microstructure [35]. In the present study, vessels located in the capsule (with a thickness ranging from 10 to 100 μm), where the transplanted islets were directly attached, were evaluated. We recently reported that constructed vessels in the subcutaneous space could be clearly observed, and these results were consistent with those of vWF staining (manuscript submitted). In the present study, the subcutaneous vessel density was significantly higher in the GHNF group in comparison to the control group in both vWF staining and lectin angiography (Figure 4B and Figure 7B). From these results, the effectiveness of lectin angiography in a rat model was confirmed, and a high number of new vessels appeared to result in good outcomes in the GHNF group (Figure 2B).

Although GHNF pretreatment was effective for improving the engraftment of subcutaneously transplanted islets in a rat model, substantial amounts of islets (5,400 IEQs), which correspond to approximately four times higher doses than intraportal transplantation [36], are still required to cure a diabetic recipient in this model. In contrast, in a mouse model, the outcome of subcutaneous islet transplantation was superior to that of intraportal islet transplantation [14]. This discrepancy may suggest that neovascularization and/or ECM compensation are still insufficient; otherwise, crucial factors, rather than neovascularization and ECM compensation, may exist in a rat model. Hence, the combination of GHNF with adipose-derived stem cells (ADSCs), which secrete various growth factors and effectively induce neovascularization [37,38,39,40], would be one option.

## 5. Conclusions

The present study showed that pretreatment using GHNF effectively improved the outcomes of subcutaneous islet transplantation in a rat model. The main mechanisms of this beneficial effect may be the induction of sufficient new vessels in the subcutaneous capsules and compensation of the ECM surrounding the transplanted islets.

## Figures and Tables

**Figure 1 cells-13-00051-f001:**
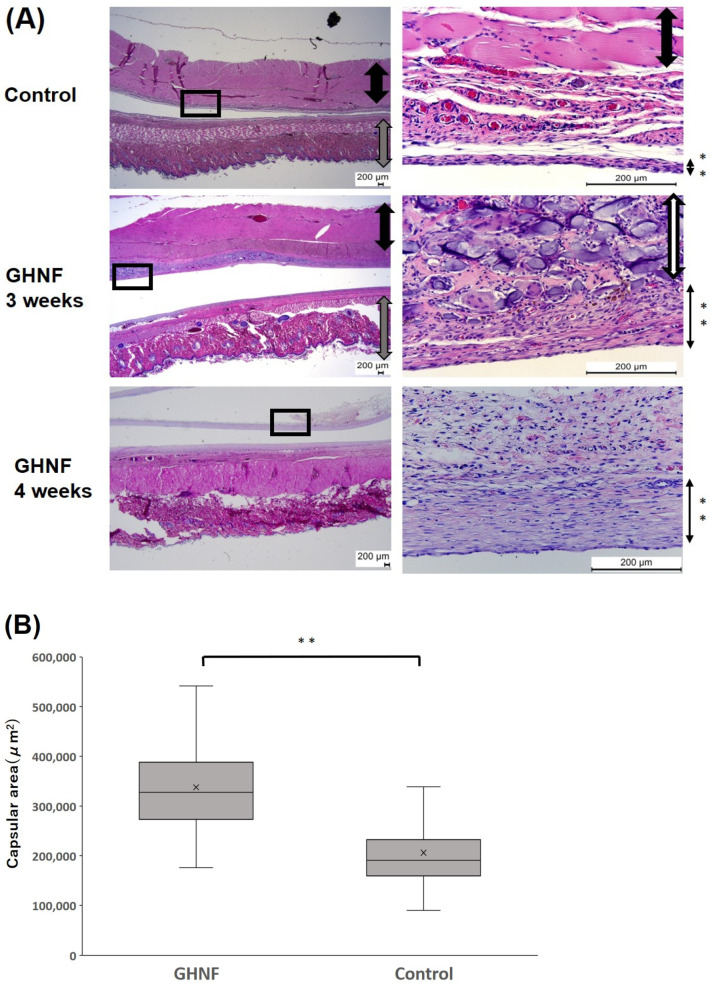
(**A**) Hematoxylin–eosin staining of the subcutaneous tissues at three and four weeks after implantation. GHNF was absorbed and its volume gradually diminished time-dependently. The pictures on the right represented the enlargement of the boxed area in the pictures on the left. Black arrow: muscles, gray arrow: skin and subcutaneous tissues, white arrow: GHNF, black arrow with **: capsules. (**B**) The area of the capsule touching the silicone spacer in the GHNF group was larger than that of the control group (**, *p* < 0.01).

**Figure 2 cells-13-00051-f002:**
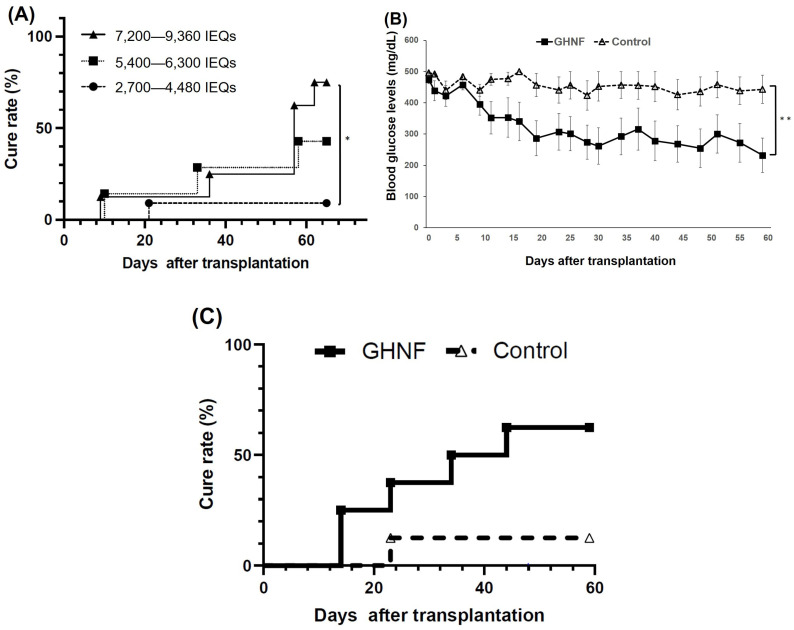
The outcomes of islet engraftment after subcutaneous islet transplantation under several conditions. (**A**) The cure rate after transplantation of different numbers of islets in the GHNF group. The cure rate of the recipients’ dose-dependently increased (2,700–4,680 IEQs [filled circle, *n* = 11]; 9.1%, 5,400–6,300 IEQs [filled square, *n* = 7]; 42%, 7,200–9,360 IEQs [filled triangle, *n* = 8]; 75%). The outcomes of islet engraftment after marginal islet mass transplantation (5,400 IEQs). (**B**) The change of the blood glucose levels after islet transplantation. GHNF (filled square, *n* = 8) group showed significantly better glucose changes than the control group (open triangle, *n* = 8) (**, *p* < 0.01). (**C**) The cure rate after islet transplantation in each group. The cure rate at 60 days after islet transplantation in the GHNF group (62.5%) was significantly higher than that in the control group (12.5%) (*; *p* < 0.05).

**Figure 3 cells-13-00051-f003:**
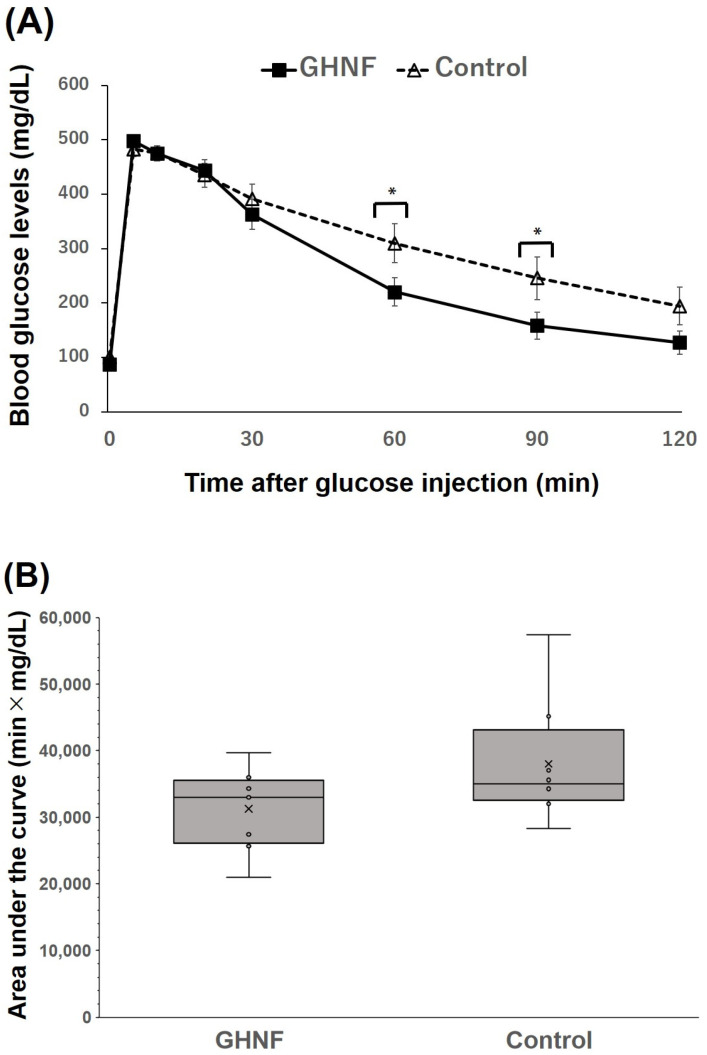
The glucose tolerance profiles of the GHNF and control groups. (**A**) Blood glucose changes in the intravenous glucose tolerance test (IVGTT) at approximately 60 days after islet transplantation. There were significant differences at 60 and 90 min after glucose injection in the IVGTT between the GHNF (filled square, *n* = 8) and control (open triangle, *n* = 8) groups (*; *p* < 0.05). (**B**) The area under the curve (AUC) of the IVGTT in each group is shown. Although the difference did not reach statistical significance, the AUC of the GHNF group was lower than that of the control group (*p* = 0.106). Each circle represents the individual data.

**Figure 4 cells-13-00051-f004:**
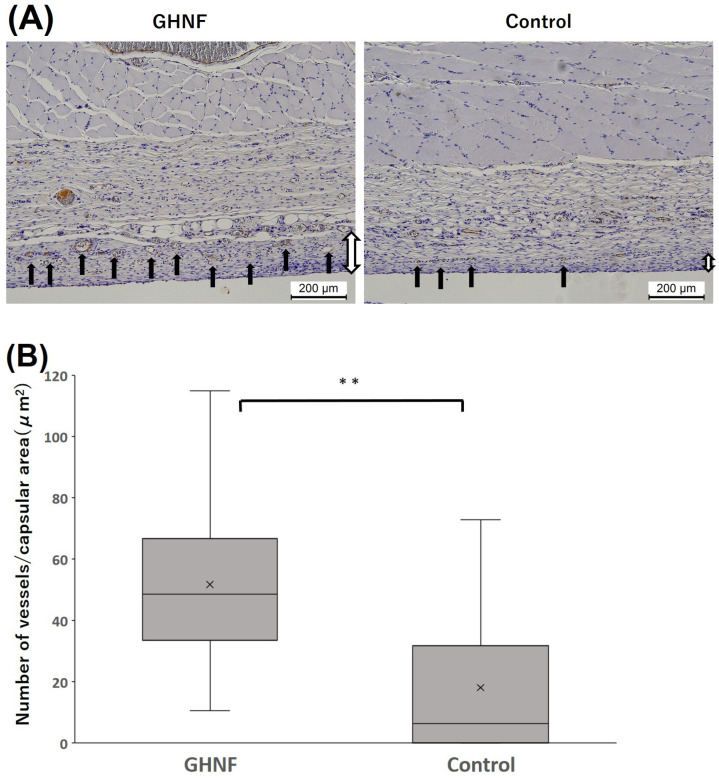
Immunohistochemical analyses of von Willebrand factor (vWF)-positive vessels. (**A**) Photomicrographs of vWF staining before islet transplantation. The vWF-positive vessels (black arrows) in the capsule around the GHNF were counted. Magnification: ×100. Calibration bars: 200 µm. (**B**) The number of vessels divided by the capsular area in each group. The number of vWF-positive vessels of the GHNF group was significantly higher than that of the control group (**, *p* < 0.01). Black arrow: vWF-positive vessels, white arrow: capsules.

**Figure 5 cells-13-00051-f005:**
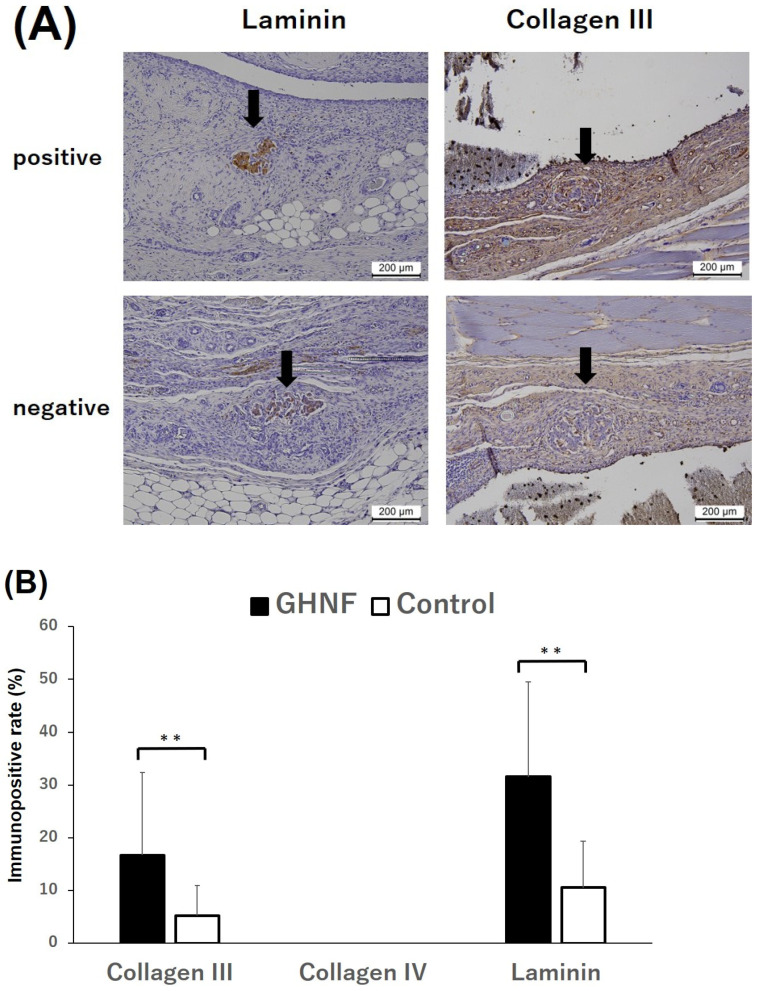
Immunohistochemical analyses of the extracellular matrix (ECM). (**A**) Representative photomicrographs of laminin and collagen III staining. Black arrows represent the transplanted islets. “Positive” for laminin, collagen III, and collagen IV indicates that distinct immunopositivity was detectable in the fibrous capsule around the islets. “Negative” indicates that immunopositivity was undetectable. Magnification: ×200. Calibration bars: 200 µm. (**B**) The rates of laminin and collagen III positivity in the GHNF (black box) and control (white box) groups. The positivity of collagen III and laminin in the GHNF group was significantly higher than that in the control group (**, *p* < 0.01).

**Figure 6 cells-13-00051-f006:**
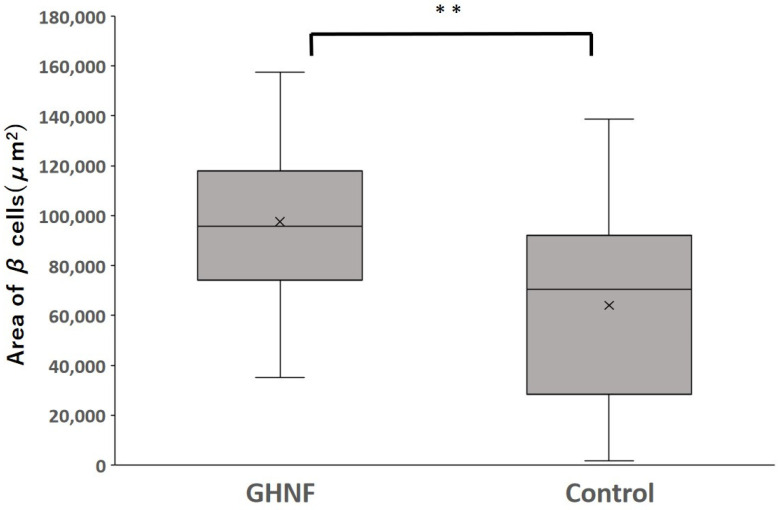
The average area of β cells in each group. The average area of β cells in the GHNF group was significantly larger than that in the control group (GHNF: 97,486 ± 4,214 vs. control: 64,022 ± 4,954 μm^2^, respectively, ** *p* < 0.01).

**Figure 7 cells-13-00051-f007:**
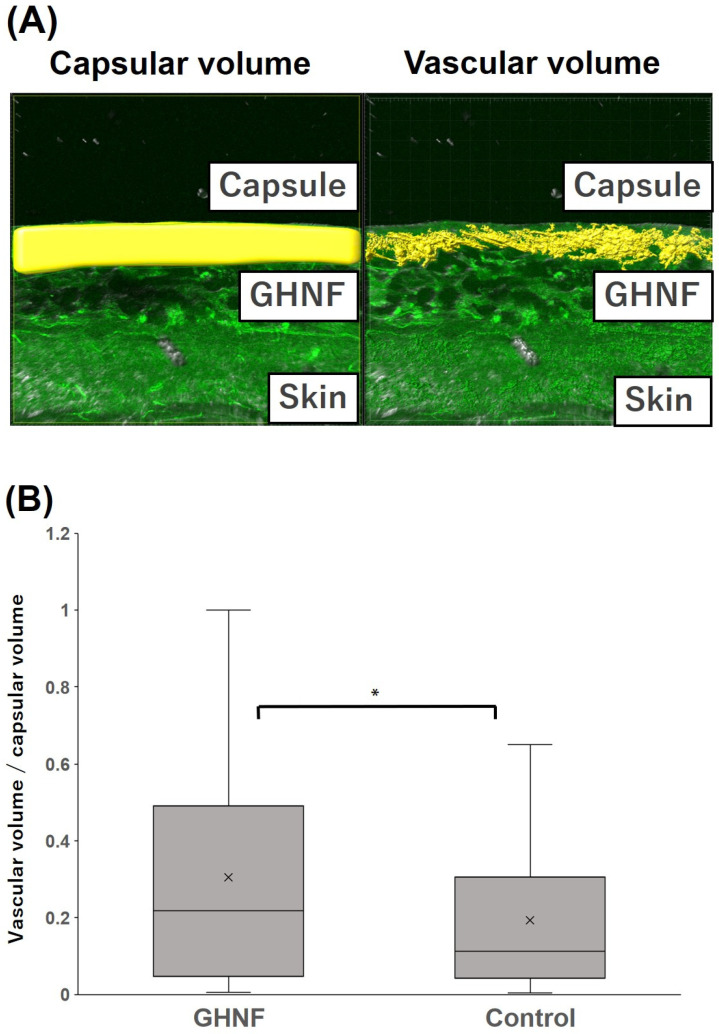
Quantification of the vascular volume analyzed using lectin angiography in the capsule at the transplant site of the GHNF and control groups. (**A**) Image of lectin angiography in the GHNF group. The volume of the capsules (left) and constructed vessels (right) (shown as a yellow structure). (**B**) The vascular density in each group. The results represent the vascular volume (μm^3^)/capsular volume (μm^3^) and the density of the vessels in the GHNF group was significantly higher than that in the control group (*, *p* < 0.05).

## Data Availability

All data generated or analyzed in this study are included in this manuscript.

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
