# Peer review of "A Gelatin Hydrogel Nonwoven Fabric Enhances Subcutaneous Islet Engraftment in Rats"

_cells, 2023, doi:10.3390/cells13010051_

Round 1

Reviewer 1 Report

Comments and Suggestions for Authors

Saito et al., demonstrated the use of gelatin hydrogel nonwoven fabric (GHFN) for subcutaneous islet transplantation in rat models. However, the results shown were insufficient, requiring additional experiments to support their claim.      

The islet dose used in the clinical trial was 5000 IEQ/kg of body weight. What was the dose of islets used in this study? Did you calculate the islet dose/cm2 area? Delivering higher doses of islets to the device leads to hypoxia, resulting in poor islet engraftment and function. In the current study, the observed diabetic cure rate was approximately 60% with gelatin hydrogel nonwoven fabric (GHFN), compared to the control (10%). To better understand the effective role of GHFN, experimentation with various islet seeding is warranted.

In your previous study (https://journals.sagepub.com/doi/epub/10.1177/09636897231186063), it was determined that the optimal time for islet delivery was 6 weeks with GHFN. However, in the current study, islets were delivered at the 3 weeks. Why?

IVGTT data did not reveal significance between the GHFN and control group, even though the recipients in the control group did not normalize. Why? To better understand the results, it is recommended to present a graph illustrating individual recipients’ data points.

In Figure 4B, are there significant differences between the groups? I noticed considerable variations in the number of vWF-positive vessels within the group. Why?    

Author Response

The islet dose used in the clinical trial was 5000 IEQ/kg of body weight. What was the dose of islets used in this study?

In this study, the islet dose was 21600 IEQ/kg of body weight.

Did you calculate the islet dose/cm2 area?

The area of GHNF sheets was about 7 cm2 when they were implanted to the subcutaneous tissue, but the transplant space may have reduced due to GHNF absorption and/or local inflammation when the islets were transplanted. Thus, I would speculate that the islet density at the transplant site might be equal to or higher than

771 IEQ/cm2. 

Delivering higher doses of islets to the device leads to hypoxia, resulting in poor islet engraftment and function. In the current study, the observed diabetic cure rate was approximately 60% with gelatin hydrogel nonwoven fabric (GHFN), compared to the control (10%).

To better understand the effective role of GHFN, experimentation with various islet seeding is warranted.

Thank you for your important comment.

Before this experiment, we compared the transplant efficiency of different islet doses to confirm the marginal graft dose in this model. As expected, the cure rate increased dose-dependently. ï½›2700-4680 IEQs (Cure rate; 9.1%, n=11), 5400-6300 IEQs (Cure rate; 42%, n=7), 7200-9360 IEQs (Cure rate; 75%, n=8)}. According to this preliminary trial, we decided to transplant 5400 IEQs of islets in the present study.

In your previous study (https://journals.sagepub.com/doi/epub/10.1177/09636897231186063), it was determined that the optimal time for islet delivery was 6 weeks with GHFN. However, in the current study, islets were delivered at the 3 weeks. Why?

Thank you very much for your insightful comment.

As we mentioned in the discussion section, the absorption rate of GHNF widely varies among animal species. In mice model, when the volume of GHNF dramatically decreased (8 weeks after implantation), the cure rate also dramatically decreased compared with 6 weeks group. From this view, GHNF should be absorbed to some extents, but should be exist to some extents for islet engraftment as a scaffold. In rat model, unlike mice model, a strong subcutaneous edema was observed at 1-2 weeks after GHNF implantation most likely due to the inflammation and there were few GHNF remained in the subcutaneous space even after 4 weeks. I also added the 4 weeks subcutaneous pictures in Fig.1A. That’s why we chose 3 weeks as an ideal duration of pretreatment in this model.

Of a particular note, we have also used the GHNF for hepatocyte subcutaneous transplantation in another study. In this study, the blood albumin level was maximized when hepatocytes were transplanted to the subcutaneous space at 3 weeks after GHNF implantation.

IVGTT data did not reveal significance between the GHFN and control group, even though the recipients in the control group did not normalize. Why? To better understand the results, it is recommended to present a graph illustrating individual recipients’ data points.

Thank you for your helpful advice.

Although there was no significant difference as a whole (p=0.16), there were  significant differences at 60 min and 90 min after glucose injection, so I added the ※ in the Figure 3A according to the reviewer’s advice. In this experiment, we could measure 500 mg/dL as the maximal glucose level due to the technical limitation of our glucometer. So, there were no differences of the glucose levels between the groups just after glucose injection (5, 10, 20, 30 min). That is the one of the biggest reasons why there was no significant difference as a whole. However, the decrease of the glucose level was much faster in the GHNF group, suggesting that the glucose tolerance in the GHNF group was definitely superior to that in the Control group.

In Figure 4B, are there significant differences between the groups? I noticed considerable variations in the number of vWF-positive vessels within the group. Why?    

Thank you for your insightful comment.

Four locations per each section were evaluated by a pathologist with a blind method and the number of vessels was evaluated. The vessels were evaluated in 120 locations in each group. The absorption degree of the GHNF differed in the location and the thickness of the membrane should have been strongly influenced by it. Vessel density was calculated by the number of vessels divided by the membrane area. The number of vessels and the area of the membrane were completely different in each location of the tissues, which was randomly selected by a pathologist. That is why there were substantial variations within the groups. Although there were variations, the average vessel density in the GHNF group was apparently much higher than that of the Control group (GHNF; 51.7 vs. Control; 18.1). According to the reviewer’s comment, we have carefully performed statistical analyses again, and confirmed that the difference in Figure 4B was statistically significant.

Reviewer 2 Report

Comments and Suggestions for Authors

In a rat model, the authors used the gelatin hydrogel nonwoven fabric (GHNF) pretreatment method for improving subcutaneous islet engraftment. The GHNF could induce efficient new vessels. They also discussed that GHNF pretreatment was effective due to the possible mechanism of neovascularization and compensation of the extracellular matrices. This reviewer raises several issues to be addressed for improving the manuscript.

1. In the abstract section, the authors don't provide objective context on the relevance and field of implementation. It should be clearer on the significant outcome.

2. In the result section, Line 197, “GHNF was absorbed, and its volume gradually decreased”. The volume of GHNF should be mentioned and not just compared with the control group.

3. Statistical test is missing in Figure 3.

4. The statistical tests in Figures 1B, 4B, 6B, and 7B should be checked. Which statistical tests were performed? Between the GHNF and control group, there is a significant overlap of the error bars, and it's hard to believe there is a statistical significance. What was the number of replicates? (N)

5.  There are many grammar concerns, for example

Line 33, “The expression of laminin and collagen III around the transplanted islets were also higher in the GHNF group (p<0.01)”. It seems that the verb “were” does not agree with the subject. Consider changing the verb form.

Line 41, It appears that you have an unnecessary comma before the dependent clause marker “because”. Consider removing the comma.

Line 49, The subordinate phrase “To solve this situation” does not appear to be modifying the subject “alternative strategies for subcutaneous islet transplantation using a two-step approach”. Rewrite the sentence to avoid a dangling modifier.

Line 72, “therefore, it can be an ideal pretreatment material for our purpose, since there is no need for removal.” it seems that you have an unnecessary comma before “since”. Consider removing the comma.

……There are still many grammar errors that need to be checked carefully.

Please uniform the format of the whole manuscript. Line 364, “6. Patents”?

Comments on the Quality of English Language

Please carefully revise the document for English language editing.

Author Response

  1. In the abstract section, the authors don't provide objective context on the relevance and field of implementation. It should be clearer on the significant outcome.

Thank you for your comment.

I added the below sentences in the abstract. (L20-25)

“the subcutaneous space is poor in vessels and transplant efficiency is still low in animal models, except in mice. Subcutaneous islet transplantation using a two-step approach have been proposed, in which a favorable cavity for transplantation is first prepared using various materials, followed by islet transplantation into the preformed cavity. We previously reported the efficacy of pretreatment using gelatin hydrogel nonwoven fabric (GHNF) and the duration of pretreatment influenced the outcomes in a mouse model.”

  1. In the result section, Line 197, “GHNF was absorbed, and its volume gradually decreased”. The volume of GHNF should be mentioned and not just compared with the control group.

Thank you for your helpful advice.

Unfortunately, we don’t have pathological samples at 1 and 2 weeks after GHNF implantation. But we have the samples at 4 weeks. So, I will add the pictures of 4 weeks sample in Figure 1A. According to the changes of the Figure, I also changed the figure legend of Figure 1A. No GHNF was observed at 4 weeks after implantation. It is difficult to calculate the GHNF volume because the absorption rate of GHNF differs in the different places. (L203-204), (L211)

  1. Statistical test is missing in Figure 3.

Although there was no significant difference as a whole (p=0.16), there were significant differences at 60 and 90 minutes after glucose injection, so I added the ※ in the Figure according to the reviewer’s advice. I also changed the result section and figure legend of Figure 3 accordingly. (L233-235), (L242-243)

  1. The statistical tests in Figures 1B, 4B, 6B, and 7B should be checked. Which statistical tests were performed? Between the GHNF and control group, there is a significant overlap of the error bars, and it's hard to believe there is a statistical significance. What was the number of replicates? (N)

Student’s t-test was performed for all these Figures.

For Figure 1B, four locations per section were calculated by a pathologist with a blind method. The thickness of the membrane was evaluated in 120 locations in each group (four locations in each slice, 6 slices, n=5 in each group). The thickness of the membrane in the GHNF group differs largely in location, that mostly depends on the volume of the remnant GHNF. This may at least in part explain why there were some overlaps of the error bars.

Figure 4B, four locations per section were evaluated by a pathologist with a blind method and the number of vessels was evaluated. The vessels were evaluated in 120 locations in each group (four locations in each slice, 6 slices, n=5 in each group). The absorption degree of the GHNF differed in the location and the thickness of the membrane should have been strongly influenced by it. Vessel density was calculated by the number of vessels divided by the membrane area. The number of vessels and the area of the membrane were completely different in each location of the tissues, which was randomly selected by a pathologist. That is why there were substantial variations within the groups. Although there were variations, the average vessel density in the GHNF group was apparently much higher than that of the Control group (GHNF; 51.7 vs. Control; 18.1).

Figure 6, total 60 islets were evaluated in the GHNF group and 54 islets were evaluated in the Control group (n=5 in each group). The average area of β cells in the GHNF group was about 1.5 times larger than that of the Control group, and this difference was confirmed to be statistically significant.

Figure 7B, four locations per section (3 slices, GHNF; n=5, Control; n=4) were evaluated and total 48 locations in the Control group and 60 locations in the GHNF group were evaluated.

Regarding all figures pointed out by the Reviewer 2, we have carefully performed statistical analyses again, and confirmed that all differences were statistically significant.

  1. There are many grammar concerns, for example

Line 33, “The expression of laminin and collagen III around the transplanted islets were also higher in the GHNF group (p<0.01)”. It seems that the verb “were” does not agree with the subject. Consider changing the verb form.

Thank you for your useful advice.

I changed “were” to “was”. (L36)

Line 41, It appears that you have an unnecessary comma before the dependent clause marker “because”. Consider removing the comma.

I deleted the comma before “because”. (L44)

Line 49, The subordinate phrase “To solve this situation” does not appear to be modifying the subject “alternative strategies for subcutaneous islet transplantation using a two-step approach”. Rewrite the sentence to avoid a dangling modifier.

To avoid a dangling modifier, I deleted “To solve this situation”. (L52)

Line 72, “therefore, it can be an ideal pretreatment material for our purpose, since there is no need for removal.” it seems that you have an unnecessary comma before “since”. Consider removing the comma.

I deleted the comma before since. (L75)

……There are still many grammar errors that need to be checked carefully.

Please uniform the format of the whole manuscript. Line 364, “6. Patents”?

I deleted the patents section. (L371-373)

Round 2

Reviewer 1 Report

Comments and Suggestions for Authors

Comments to the Author:

As previously mentioned, to better understand the effective role of GHFN, experimentation with various islet seeding is warranted. Since you have already conducted this experiment, it is necessary to include this data in the revised manuscript. Incorporating this information will significantly contribute to the study and provide valuable insights to the audience.

I inquired about the dose islet dose used in this study, and you mentioned it was 21600 IEQ/kg of body weight. This appears to be four times higher than the clinical dose. Could you please provide insights into why such a significantly higher dose was necessary?

As previously mentioned, it is essential to include a graph in Figure 3B illustrating individual recipients’ data points.

Author Response

As previously mentioned, to better understand the effective role of GHFN, experimentation with various islet seeding is warranted. Since you have already conducted this experiment, it is necessary to include this data in the revised manuscript. Incorporating this information will significantly contribute to the study and provide valuable insights to the audience.

Thank you for your valuable comment. I added a new figure of the result after different dose of islet transplantation using GHNF as Figure 2A. With this change, I added some sentences in the method section (L128-132), results section (L222-227) and figure legend (L237-240).

I inquired about the dose islet dose used in this study, and you mentioned it was 21600 IEQ/kg of body weight. This appears to be four times higher than the clinical dose. Could you please provide insights into why such a significantly higher dose was necessary?

Thank you for your insightful comment. In a mouse model using GHNF, we could achieve normoglycemia with only about 200 IEQs (correspond to approximately 8000 IEQs/kg) of islets, and that result was superior to the intraportal transplantation, that is the current standard procedure for humans. Other article also mentioned that 250 IEQs (approximately 10000 IEQs/kg) of islets cured the diabetic mice when they were transplanted into the pretreated subcutaneous space (Biomaterials,2021:269,120499). However, unlike mouse model, they needed to transplant 4000 IEQs (approximately 16000 IEQs/kg) of islets to cure diabetic rats by co-transplanting with collagen and human umbilical vein endothelial cells (Biomaterials,2021:269,120499,). Likewise, several previous studies have reported that 2500 to 5000 IEQs (approximately 10000 to 20000 IEQs/kg) of islets were needed to cure diabetic rats by performing subcutaneous islet transplantation even under the preferable pretreatments (Transplantation, 2006:81:1318-1324, Cell Transplantation, 2005:14:595-605, Pancreas, 2001:23:375-381). From this point of view, our outcomes were consistent with them and the subcutaneous environment of rats is much more severe than that of mice.

In another respect, approximately 10000 to 12000 IEQs/kg of islets were required to achieve insulin independence even in the intraportal transplantation. Given that subcutaneous islet transplantation generally requires two or three times-higher amounts of islets to cure diabetic patients, we may speculate that the outcome of our present study (21600 IEQs/kg) is logical, otherwise within the scope of the assumption.

However, some previous researchers have already succeeded to cure diabetic rats by transplanting smaller amounts of islets even in a subcutaneous transplantation model. Therefore, we definitely need to further refine our approach, as the Reviewer 1 pointed out. GHNF was originally constructed for the cell culture, especially aiming for mesenchymal stem cells (MSCs). Therefore, a strong impact would be expected if GHNF is combined with MSCs. In fact, this combination significantly induced larger numbers of vessels and improved the subcutaneous islet transplantation results in mice model (article submitted). Although the impact of GHNF alone in a rat model is limited, we believe that GHNF is an ideal material when they are combined with several types of cells and/or some kind of growth factors. Therefore, this article is an essential first step for further refinement of our approach.

As previously mentioned, it is essential to include a graph in Figure 3B illustrating individual recipients’ data points.

Thank you for your comment. I added each individual data in Figure 3B.

Reviewer 2 Report

Comments and Suggestions for Authors

NONE 

Author Response

Thank you for your comment.